# Gene Network Analysis of the Transcriptome Impact of SARS-CoV-2 Interacting MicroRNAs in COVID-19 Disease

**DOI:** 10.3390/ijms23169239

**Published:** 2022-08-17

**Authors:** Alexandra Ioana Moatar, Aimee Rodica Chis, Catalin Marian, Ioan-Ovidiu Sirbu

**Affiliations:** 1Department of Biochemistry and Pharmacology, Discipline of Biochemistry, University of Medicine and Pharmacy “Victor Babes”, E. Murgu Square No. 2, 300041 Timisoara, Romania; 2Doctoral School, University of Medicine and Pharmacy “Victor Babes”, E. Murgu Square No. 2, 300041 Timisoara, Romania; 3Center for Complex Network Science, University of Medicine and Pharmacy “Victor Babes”, E. Murgu Square No. 2, 300041 Timisoara, Romania

**Keywords:** microRNA, SARS-CoV-2, network analysis

## Abstract

According to the World Health Organization (WHO), as of June 2022, over 536 million confirmed COVID-19 disease cases and over 6.3 million deaths had been globally reported. COVID-19 is a multiorgan disease involving multiple intricated pathological mechanisms translated into clinical, biochemical, and molecular changes, including microRNAs. MicroRNAs are essential post-transcriptional regulators of gene expression, being involved in the modulation of most biological processes. In this study, we characterized the biological impact of SARS-CoV-2 interacting microRNAs differentially expressed in COVID-19 disease by analyzing their impact on five distinct tissue transcriptomes. To this end, we identified the microRNAs’ predicted targets within the list of differentially expressed genes (DEGs) in tissues affected by high loads of SARS-CoV-2 virus. Next, we submitted the tissue-specific lists of the predicted microRNA-targeted DEGs to gene network functional enrichment analysis. Our data show that the upregulated microRNAs control processes such as mitochondrial respiration and cytokine and cell surface receptor signaling pathways in the heart, lymph node, and kidneys. In contrast, downregulated microRNAs are primarily involved in processes related to the mitotic cell cycle in the heart, lung, and kidneys. Our study provides the first exploratory, systematic look into the biological impact of the microRNAs associated with COVID-19, providing a new perspective for understanding its multiorgan physiopathology.

## 1. Introduction

The late 2019 SARS-CoV-2 infections reported in Wuhan province rapidly spread worldwide due to their high transmissibility and the lack of appropriate, coordinated prompt public health measures [1]. The World Health Organization (WHO) has reported over 536 million confirmed cases globally as of June 2022 [2]. Although initially viewed as primarily a respiratory disease, accumulating evidence showed that COVID-19 is a complex, multisystem disorder in patients with mild-to-severe clinical presentations [3,4].

The multiorgan (lung, heart, kidney, and liver) failure associated with high-load SARS-CoV-2 infection is thought to be a consequence of the quasi-ubiquitous expression of the ACE2 receptor (the main entry point of the virus in human host cells) [5]. In these tissues, the interaction between the ACE2 receptor and viral spike protein S promotes the unbalanced release of proinflammatory cytokines, leading to a cascade of pathological changes, including endothelial dysfunction, coagulation abnormalities, and massive infiltration of inflammatory cells [6,7]. These changes evolve in tissue-specific damage such as acute respiratory distress syndrome, pulmonary embolism, myocardial inflammation and arrhythmias, acute liver and kidney injury, and metabolic encephalopathy [4].

High-throughput profiling showed that these tissue-specific damages are associated with particular transcriptional profiles in each of the tissues infected with SARS-CoV-2. Park et al. characterized the body-wide transcriptome changes in relation to SARS-CoV-2 load and described coordinated organ-specific alterations of multiple signaling pathways in the heart, lung, and lymph nodes [8]. In another study, Delorey et al. analyzed the transcriptional profiles of several tissues such as the lung, kidney, liver, and heart. They described extensive tissue-specific transcriptional alterations underlying the pathophysiology of multiple organ failures associated with COVID-19 [9].

Transcriptome and (coding and noncoding) gene network analyses are paramount to understanding disease pathophysiology [10]. MicroRNAs are small noncoding RNAs that regulate gene expression by interacting with and modulating target RNAs’ stability and translation. The mRNA–microRNA interaction triggers repression of translation and/or degradation of target mRNA; extensive microRNA–mRNA complementarity can also lead to microRNA degradation through a target RNA-directed microRNA degradation mechanism [11]. The impressive number of microRNAs abnormally expressed in cancer, multiple organ diseases, and immune disorders highlights their relevance to human pathologies, including infectious diseases [12]. The pleiotropic roles of microRNAs in infectious diseases range from regulating leukocyte response to directly targeting the viral transcripts, thus facilitating pathogen clearance [13].

Like other RNA virus infections, the relationship between host microRNAs and SARS-CoV-2 is biunivocal. On the one hand, host microRNAs might directly inhibit viral replication, block cellular receptors, and obstruct the functioning of viral proteins, thus hindering the development of the disease [14,15,16,17]. On the other hand, infection with SARS-CoV-2 induces extensive changes in the host miRnome [18,19]. Many bioinformatic studies have used target prediction algorithms to identify the host microRNA interacting with the SARS-CoV-2 genome (Appendix A). Most of the published experimental studies have investigated the expression of tissue or circulant human microRNAs upon SARS-CoV-2 infection to validate novel diagnostic or prognostic COVID-19 biomarkers and elucidate various aspects of COVID-19 physiopathology. Nevertheless, little is known about the collective impact of the predicted and experimentally validated SARS-CoV-2 interacting host microRNAs on the multiorgan transcriptomic profiles in COVID-19 disease.

Our study had two aims. First, we were interested to see which microRNAs predicted to interact with the SARS-CoV-2 genome have experimentally been validated as differentially expressed in COVID-19 patients. Second, we used gene network analysis to evaluate these microRNAs’ transcriptomic impact and functionally characterize the microRNA-induced tissue-specific responses in COVID-19.

## 2. Results

Upon abstract collection, 44 research papers (23 in silico [20,21,22,23,24,25,26,27,28,29,30,31,32,33,34,35,36,37,38,39,40,41,42] and 21 experimental studies [43,44,45,46,47,48,49,50,51,52,53,54,55,56,57,58,59,60]) met the inclusion criteria and were used for downstream analysis. The entire workflow of the current study is depicted in Figure 1.

Overall, the in-silico prediction approaches identified 756 mature microRNAs targeting SARS-CoV-2 compared with 164 microRNAs from experimental studies. Of note, some of these microRNAs also target *ACE2*, *TMPRSS2*, *IFN-alpha*, *IFN-beta*, and *IFN-gamma*. Only 54 (26 downregulated and 28 upregulated, Table 1) of the predicted microRNAs were experimentally validated as differentially expressed in COVID-19 patients.

MicroRNAs target prediction (miRWalk 3.0, binding probability > 0.95) identified 13,268 unique targets for the downregulated set of microRNAs and 13,978 unique targets for the upregulated ones (Appendix A).

Integrative analysis of multiple transcriptome datasets is challenging due to data heterogeneity, missing data between datasets, and issues related to scalability and dimensionality [61]. Therefore, we cross-referenced the two predicted microRNA target lists with the unitary, comprehensive, in-depth spatial, and tissue-specific (heart, lymph node, lung, kidney, and liver) DEG of the GSE169504 dataset [8]. The list of organ-specific DEGs predicted to be microRNA targets (Appendix A) was further used for gene network functional enrichment analysis using the STRING algorithms.

The heart downregulated DEGs targeted by upregulated microRNAs formed a complex network with 1438 nodes and 4194 edges; Markov clustering analysis identified a large cluster with 134 nodes and 708 edges (protein–protein interaction enrichment, PPI = 1 × 10^−16^), the functional enrichment analysis of which (Figure 2A) highlighted a significant impact on multiple biological processes, molecular functions, and KEGG pathways. The most important terms in this cluster were the Gene Ontology (GO) biological process *mitochondrial respiratory chain complex assembly* (22 genes) and the Kyoto Encyclopedia of Genes and Genomes (KEGG) pathway *oxidative phosphorylation* (37 genes). The heart upregulated DEGs targeted by downregulated microRNAs formed a complex network with 1127 nodes and 1248 edges; Markov clustering analysis identified a large cluster with 24 nodes and 88 edges (PPI = 1 × 10^−16^), the functional enrichment analysis of which (Figure 2B) retrieved the GO biological process *mitotic cell cycle* (19 genes).

The 362 downregulated DEGs targeted by upregulated microRNAs in the lungs formed a network with 362 nodes and just 107 edges, which was unsuitable for enrichment analysis. The upregulated DEGs targeted by downregulated microRNAs formed a complex network of 584 nodes and 1993 edges. After Markov clustering, the enrichment analysis of the largest cluster (127 genes, 1348 edges, PPI = 1 × 10^−16^) identified the *cell cycle* (84 genes) as the biological process and KEGG pathway most significantly impacted by microRNAs (Figure 3).

In the lymph nodes, the 369 downregulated DEGs impacted by the upregulated microRNAs formed a network of 369 nodes and 180 edges. After Markov clustering, the largest cluster identified comprised 16 genes with 33 edges (PPI = 3.6 × 10^−6^) (Figure 4), the functional enrichment analysis of which depicted positive regulation of *cytokine production* (biological process) and *primary immunodeficiency* (KEGG pathway) as the most impacted by microRNA. The lymph nodes upregulated DEGs potentially targeted by the downregulated microRNAs formed a network with 322 nodes and only 51 edges, which was unsuitable for functional enrichment analysis.

In the kidneys, the downregulated DEGs impacted by the upregulated microRNAs formed a network with 717 genes and 421 nodes. After Markov clustering, the largest cluster contained 87 nodes and 153 edges (PPI = 1 × 10^−16^); its enrichment analysis identified the *cell-surface receptor signaling pathway* (33 genes) and KEGG pathway *proteoglycans in cancer* (14 genes) as most significantly impacted by microRNAs (Figure 5A). The upregulated DEGs targeted by downregulated microRNAs formed a complex network of 923 nodes with 1728 edges. The results of Markov analysis showed that the largest cluster contained 27 nodes with 130 edges (PPI = 1 × 10^−16^), and enrichment analysis identified the *mitotic cell cycle* (16 genes) as the most significantly impacted biological process (Figure 5B).

In the liver, the downregulated DEGs targeted by upregulated microRNAs formed a network of 304 nodes with 122 edges. After Markov clustering, the largest cluster comprised 30 nodes and 46 edges (PPI = 2.46 × 10^−7^), the enrichment analysis of which retrieved GO biological process *response to stimulus* (28 genes), GO molecular function *signaling receptor binding* (14 genes), and KEGG pathway *proteoglycans in cancer* (6 genes) as the most impacted (Figure 6A). The downregulated DEGs targeted by upregulated microRNAs formed a network of 434 nodes and 253 edges. After Markov clustering, enrichment analysis of the largest cluster (12 nodes and 14 edges, PPI = 6.09 × 10^−9^) identified the *PI3K-Akt signaling* KEGG pathway (eight genes) as significantly impacted by microRNAs (Figure 6B).

We provide all the networks generated using the STRING app in the Cytoscape platform in Appendix A.

## 3. Discussion

MicroRNAs are noncoding RNAs with a central place in all gene regulatory networks, regardless of the biological context: a single microRNA can modulate the stability of hundreds of mRNAs, and a single mRNA may interact with tens of microRNAs [62]. Therefore, the role of microRNAs in COVID-19 diagnostic, prognostic, and physiopathology has been of particular interest, with in silico and experimental data evidence accumulating at a steady pace. Both pathogenic and nonpathogenic coronaviruses have been shown to have hundreds of potential interaction sites with human microRNAs, with a possible sponging effect that can impact host tissues’ normal miRnome physiology [63,64,65]. Here, we aimed to gather all host microRNAs in silico predicted to interact with SARS-CoV-2 and experimentally validated as differentially expressed in COVID-19 and to evaluate their biological impact using gene network analysis of target DEGs from tissue-specific transcriptomes.

With the surprising exception of the lung (upregulated) and the lymph node (downregulated) microRNAs, our analyses predicted that host microRNA dysregulation triggers tissue-specific, biologically relevant transcriptomic changes in all organs with high SARS-CoV-2 viral loads, which can contribute to multiorgan failure in severe COVID-19 cases. We found that the upregulated microRNAs control processes such as mitochondrial respiration and cytokine and cell surface receptor signaling pathways in the heart, lymph node, and kidneys. In contrast, downregulated microRNAs are primarily involved in processes related to the mitotic cell cycle in the heart, lung, and kidneys.

The mitochondrial impact of SARS-CoV-2 infection is a topic gaining traction as experimental data on dysfunctional mitochondrial metabolism and transcriptome have been accumulating [66]. Cardiac cells have high ATP needs, and their activity heavily relies on mitochondrial health [67]. SARS-CoV-2 localizes in mitochondria and alters the composition of mitochondrial complex I, leading to a redox imbalance that might explain not only the acute myocardial injury but also the long COVID-19 syndrome [68,69,70]. Our heart transcriptome analysis revealed a strong, unanticipated impact of microRNAs on the mitochondrial function in COVID-19, namely the mitochondrial respiratory chain complex assembly and oxidative phosphorylation. These results are in line with Li et al.’s findings, which showed alterations in mitochondrial respiratory chain complex and oxidative phosphorylation in a hACE2/SARS-CoV-2 mouse model [71].

The network analysis of downregulated genes in COVID-19 kidneys and liver showed a significant enrichment in pathways related to proteoglycans and processes related to cell surface receptor and response to stimulus. Cell surface proteoglycans are directly involved in the attachment, internalization, and intracellular trafficking of SARS-CoV-2 [72,73,74]. The multiorgan endothelial dysfunction in severe COVID-19 cases is attributed to the ubiquitous expression of ACE2 receptors; however, this cannot be the unique factor involved in this process. It is assumed that, in addition to ACE2, other factors such as heparan sulfate proteoglycans are likely to act as coreceptors for S protein and thus contribute to multiorgan complications of COVID-19 [75]. Our data suggest that the host cell’s miRnome changes can modulate the expression of the ACE2 receptor and proteoglycan SARS-CoV-2 coreceptor.

Finally, our network analysis of downregulated genes in lymph nodes highlighted the impact of microRNAs on the processes related to the regulation of cytokine production and primary immunodeficiency. The role of cytokines in the physiopathology of COVID-19 has been widely documented, and the circulant levels of cytokines such as Il-6 and TNF-alpha are strong predictors of COVID-19 progression [76,77,78]. High SARS-CoV-2 viral loads trigger an acute inflammatory response, with the subsequent imbalanced release of pro-inflammatory molecules, leading to multiorgan damage [79,80]. Combined microRNAs and cytokine and chemokine profiling significantly improve the prediction of COVID-19 evolution to severe or fatal clinical forms, even though the microRNAs involved are not differentially expressed in severe vs. mild disease forms [81], indicating progressive recruitment of novel microRNA-modulated gene networks. Garcia-Giralt et al. found that the differentially expressed circulant microRNAs in severe (mechanically ventilated) COVID-19 patients primarily target the acute inflammatory response pathway [82]. The strong cytokine–microRNA connection opens the perspective for therapeutically controlling cytokine storm by using microRNAs to target and modulate the expression of proinflammatory molecules at the mRNA level [83].

Interestingly, cluster analysis of the transcriptomic effect of down-regulated microRNAs identified pathways and biological processes related to the mitotic cell cycle in three of the five investigated organs: heart, lung, and kidneys. The effect of viruses on the host cell cycle has been extensively researched, and several studies have reported the coronaviruses’ specific strategies to hinder particular phases of the host cell cycle to the benefit of their replication cycle. Host cells are arrested in the G0/G1 phase of the cell cycle through the combined action of SARS-CoV 3a [84], 3b [85], and 7a [86] proteins. SARS-CoV was shown to interfere with the progression of the host cell cycle through direct interaction of N protein with cyclin D or cyclin-CDK2 and Nsp13 with the p125 subunit of DNA polymerase δ [87]. Furthermore, the SARS-CoV nucleocapsid-mediated inhibition of the cyclin–cyclin-dependent kinase complex blocks the host cells in the S phase of the cell cycle [88]. Wang et al. showed that SAS-CoV-2 N protein could trigger a Smad3-dependent G1 cell-cycle arrest, thus exacerbating initial acute kidney injury and inducing the death of renal tubular epithelial cells [89].

The role of the cell cycle in the development of COVID-19 heart injury was documented in a gene set enrichment (GSE) analysis of human-induced pluripotent stem-cells-derived cardiomyocytes infected with SARS-CoV-2, which revealed a substantial reduction in the expression of genes related to G2/M checkpoint [90]. Alterations of the cell-cycle progress were also documented in lung tissues in a mouse model, where a multi-omics study showed dynamic changes in cell cycles in conjunction with alterations in the CDK and MAPK signaling networks [91].

Our liver cluster analysis identified the PI3K–Akt signaling pathway as significantly impacted by downregulated microRNA changes. The hyperactivation of the PI3K–Akt pathway in COVID-19 has been linked to SARS-CoV-2 in various ex vivo models [92,93] and was shown to mediate viral entry and induce a consistent platelet activation response [94]. Of note, both furin and CD147, known SARS-CoV-2 receptors, are expressed on platelets [95,96], providing a possible explanation for SARS-CoV-2 induced platelet activation and thrombi formation. In addition to these direct mechanisms, SARS-CoV-2 might indirectly activate the PI3K–Akt pathway through Ang II and inflammatory cytokines [97]. MicroRNA modulation of PI3K–Akt signaling provides yet another new therapeutic option.

PI3K–Akt signaling is also involved in the development of fibrotic processes [98,99] in the lungs [100,101] and liver; of note, COVID-19-related liver fibrosis can develop independently of pre-existing chronic liver disease [102]. The link between liver fibrogenesis, microRNA, and the PI3K signaling pathway was investigated; in an experimental liver fibrosis model, Lei et al. reported that miR-101 has a strong antifibrotic effect through downregulation of the PI3K-Akt signaling pathway [103].

## 4. Methods and Materials

### 4.1. Abstract Collection

We searched the PubMed repository with the terms “microRNAs AND COVID-19 disease” and “microRNA AND SARS-CoV-2 infection” for research articles published between December 2019 and December 2021. Both in silico (with or without subsequent experimental validation) and ex vivo and in vivo experimentally generated data regarding human host microRNAs were included. Books, book chapters, reviews, meta-analyses, editorials, comments, conference abstracts, and clinical trials were excluded. We also excluded all research articles analyzing SARS-CoV-2 encoded microRNAs. The entire dataset of host microRNAs retrieved from our PubMed query is presented in Appendix A.

### 4.2. Prediction of MicroRNAs’ Target Genes

The microRNA species common to both lists (experimentally validated and in silico predicted) were stratified into two groups (downregulated and upregulated) and used for subsequent analyses. MicroRNAs with conflicting experimental expressions were excluded from downstream analysis. Target predictions were computed using the miRWalk3.0 machine learning algorithm with the following criteria: (1) microRNA binding probability above 0.95; (2) Bonferroni adjusted *p*-value less than 0.05 as a cutoff; and (3) 3’UTR selected as microRNAs specific target sequence. Given that microRNAs’ binding to 3’UTR of mRNAs decreases their stability, we assumed that the predicted target genes of downregulated microRNAs are upregulated, and the target genes of upregulated microRNAs are downregulated [104].

### 4.3. Network and Functional Enrichment Analysis

The microRNAs’ target gene lists were cross-referenced with the differentially expressed gene (DEG) datasets (FDR adjusted *p*-value < 0.05) published by Park et al. in their system-wide transcriptome study of COVID-19 patients (Appendix A). The heart, lymph node, lung, liver, and kidney DEGs (vs. controls) used for cross-referencing were all retrieved from transcriptomes of patients with high viral loads of SARS-CoV-2 [8]. All tissue-specific microRNA-targeted DEGs were uploaded using the STRING app on the Cytoscape platform for gene network and functional enrichment analysis [105].

For network clustering, we used the clusterMarker2 app and the Markov clustering (MCL) method with an inflation value set to 4 (to reduce the cluster size) and using STRING interactions scores predicted with a high (0.7) confidence score. Only the largest cluster in each network, generated with the background genes set to be the String network, was further used for functional enrichment analysis.

The functional enrichment analyses of the microRNA-targeted DEGs clusters were performed using the String Enrichment app, focusing on Gene Ontology terms such as biological processes, molecular functions, and KEGG pathways. The most significant GO terms and pathways were selected using an FDR threshold of 5% and a redundancy score set at 0.5 as a cutoff to eliminate redundant terms.

## 5. Conclusions

Our data indicate that SARS-CoV-2 uses coronavirus-interacting human microRNAs to modulate host cellular processes and signaling pathways that impact COVID-19 physiopathology. The cluster analyses of the microRNA-dependent DEG networks identified mitochondrial respiration, cytokine and cell surface receptor signaling, and mitotic cell cycle as commonly regulated by microRNAs in multiple organs. From this perspective, our multi-organ transcriptome analysis is relevant for understanding COVID-19 physiopathology and narrows the spectrum of microRNA-dependent molecular mechanisms involved in COVID-19 multiorgan failure. Lastly, our data provide a rationale for designing targeted, microRNA-modulating therapeutical approaches to COVID-19.

The major limitation of our study is that it heavily relied on predicted microRNA–SARS-CoV-2 and microRNA–mRNA interaction data. Therefore, our data need extensive experimental validation, from assessing the microRNA networks in infected tissues, validating microRNAs’ interactomics, to altering specific microRNAs’ balance using microRNA analogs and inhibitors.

## Figures and Tables

**Figure 1 ijms-23-09239-f001:**
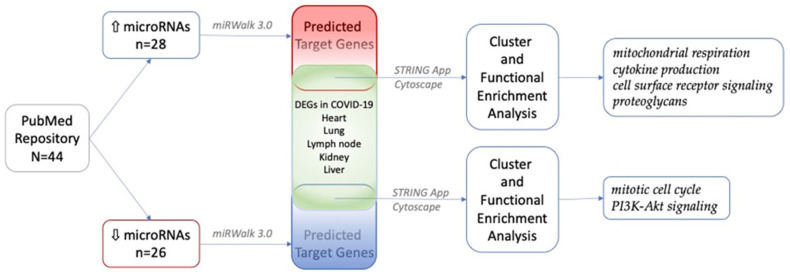
The analytical pipeline used in the current study. N—number of research papers included in the analysis; n—number of microRNAs; arrow up—upregulated microRNAs; arrow down—downregulated microRNAs.

**Figure 2 ijms-23-09239-f002:**
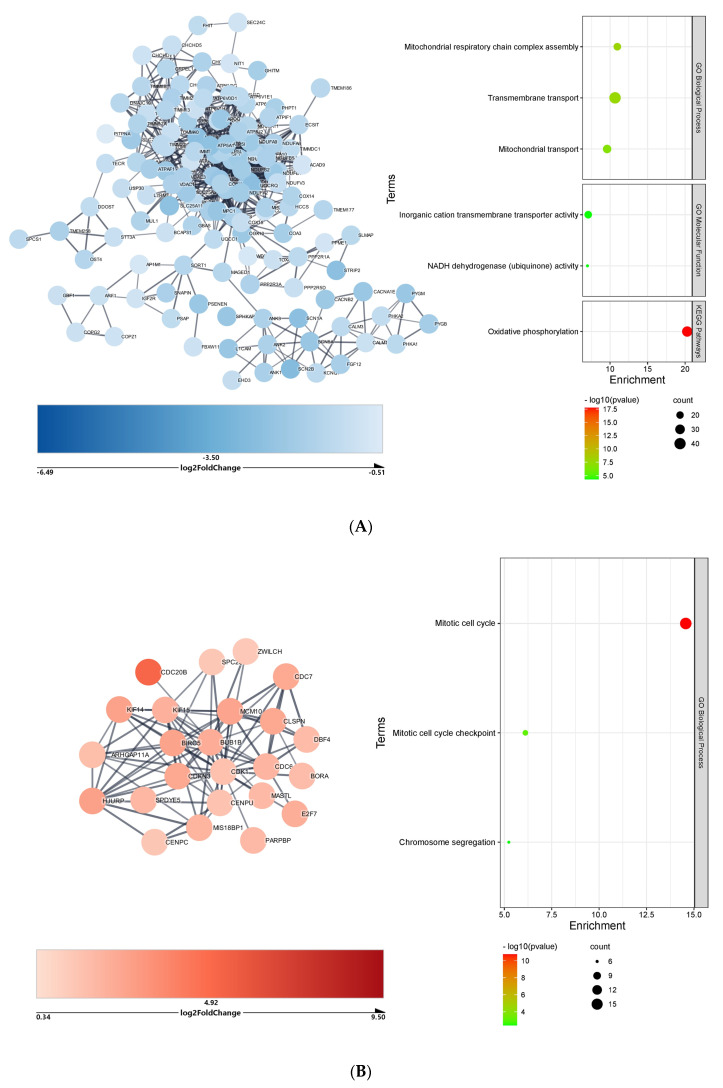
Cluster and functional enrichment analysis of COVID−19 heart DEGs targeted by differentially expressed microRNAs: STRING interaction confidence score > 0.7; Markov clustering inflation value = 4; STRING Enrichment FDR < 0.05; redundancy score = 0.5. (**A**) Cluster and functional enrichment analysis of COVID-19 heart downregulated DEGs targeted by upregulated microRNAs. (**B**) Cluster and functional enrichment analysis of COVID-19 heart upregulated DEGs targeted by downregulated microRNAs.

**Figure 3 ijms-23-09239-f003:**
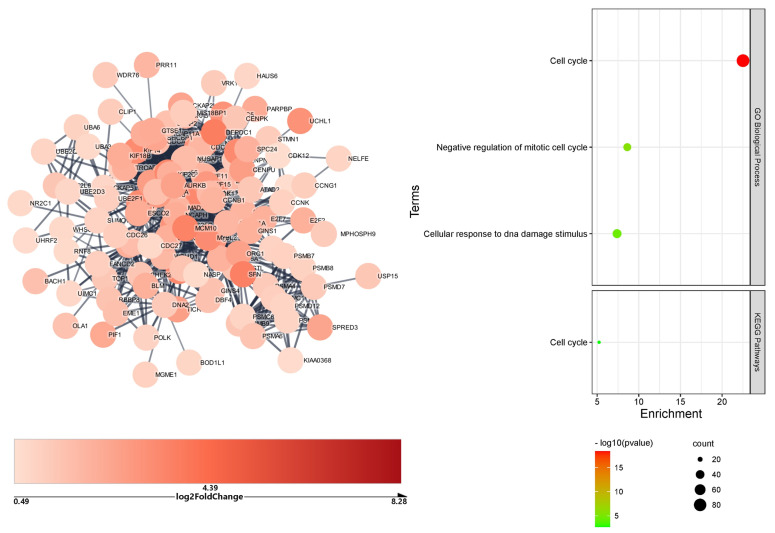
Cluster and functional enrichment analysis of COVID−19 lung upregulated DEGs targeted by downregulated microRNAs. STRING interaction confidence score > 0.7; Markov clustering inflation value = 4; STRING Enrichment FDR < 0.05; redundancy score = 0.5.

**Figure 4 ijms-23-09239-f004:**
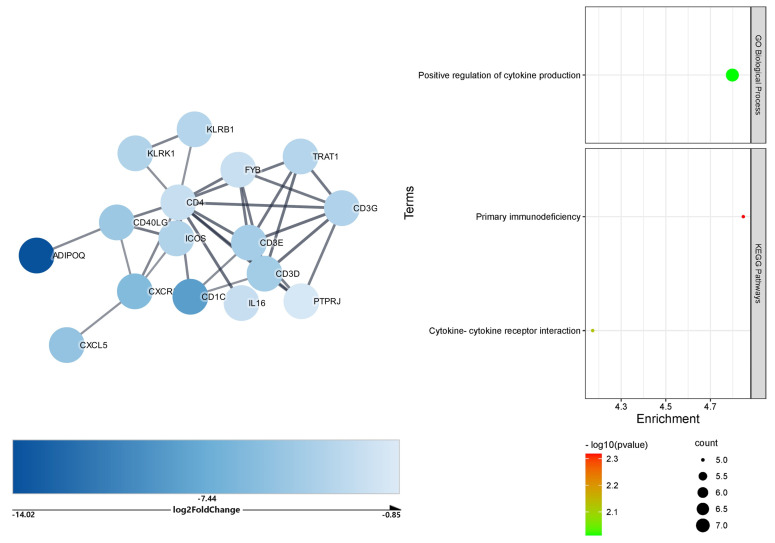
Cluster and functional enrichment analysis of COVID−19 lymph node downregulated DEGs targeted by upregulated microRNAs. STRING interaction confidence score > 0.7; Markov clustering inflation value = 4; STRING Enrichment FDR < 0.05; redundancy score = 0.5.

**Figure 5 ijms-23-09239-f005:**
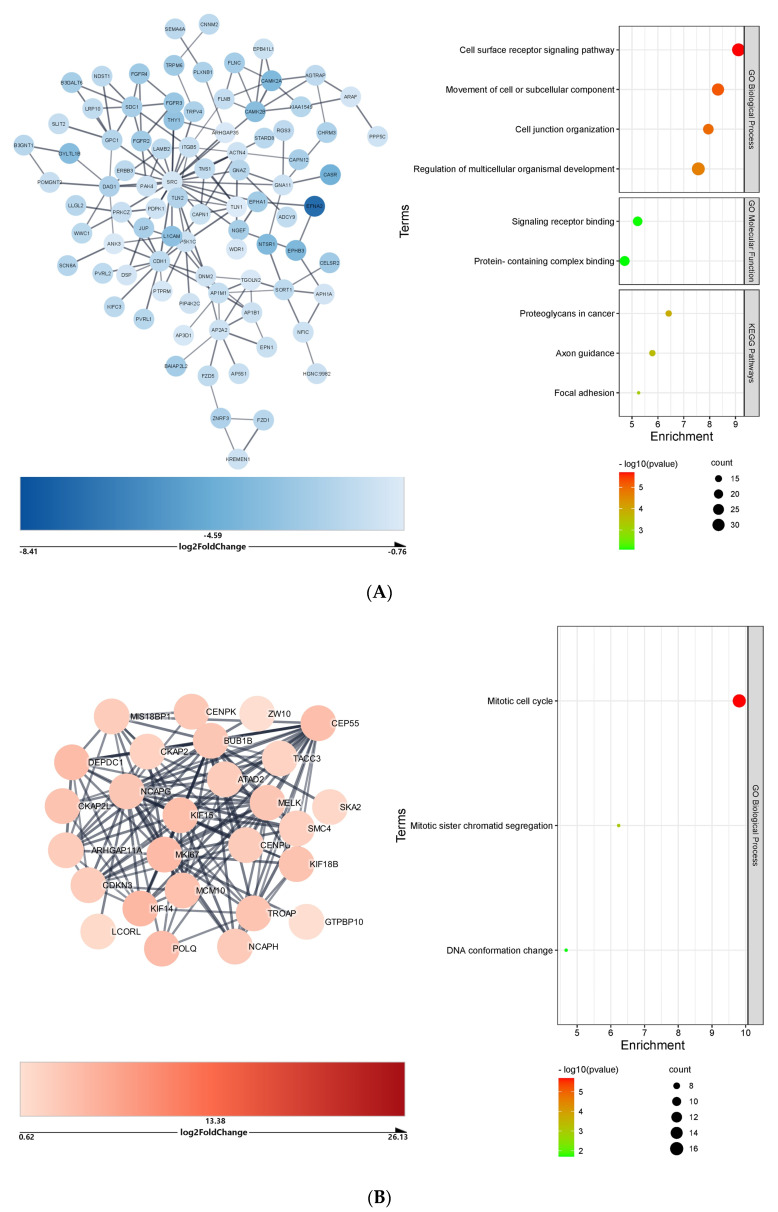
Cluster and functional enrichment analysis of COVID−19 kidneys DEGs targeted by differentially expressed microRNAs: STRING interaction confidence score > 0.7; Markov clustering inflation value = 4; STRING Enrichment FDR < 0.05; redundancy score = 0.5. (**A**) Cluster and functional enrichment analysis of COVID-19 kidney downregulated DEGs targeted by upregulated microRNAs. (**B**) Cluster and functional enrichment analysis of COVID-19 kidney upregulated DEGs targeted by downregulated microRNAs.

**Figure 6 ijms-23-09239-f006:**
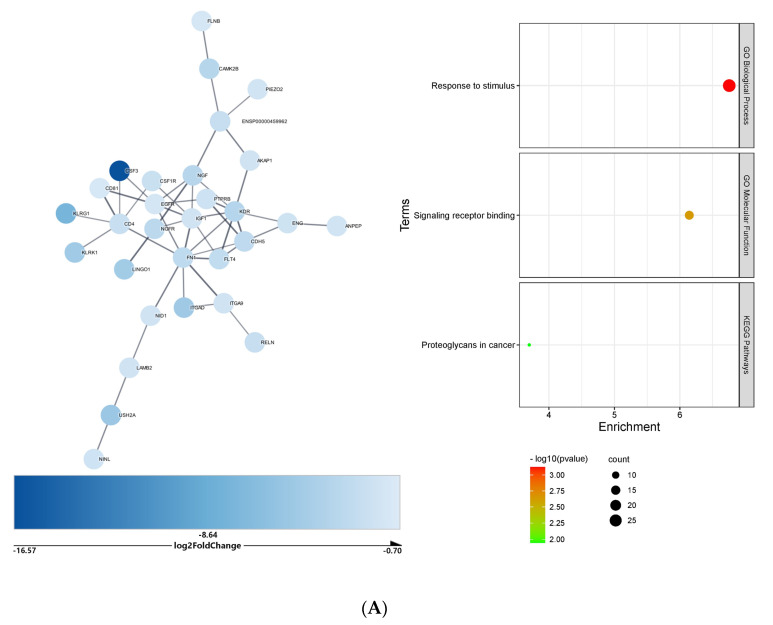
Cluster and functional enrichment analysis of COVID-19 liver DEGs targeted by differentially expressed microRNAs: STRING interaction confidence score > 0.7; Markov clustering inflation value = 4; STRING Enrichment FDR < 0.05; redundancy score = 0.5. (**A**) Cluster and functional enrichment analysis of COVID-19 liver downregulated DEGs targeted by upregulated microRNAs. (**B**) Cluster and functional enrichment analysis of COVID-19 liver upregulated DEGs targeted by downregulated microRNAs.

**Table 1 ijms-23-09239-t001:** List of miRNAs in silico predicted and experimentally validated.

Downregulated microRNAs	References	Upregulated microRNAs	References
hsa-miR-1226-3p	Farr et al., 2021 [19]	hsa-let-7e-5p	Farr et al., 2021 [19]
hsa-miR-1275		hsa-let-7f-5p	
hsa-miR-145-3p		hsa-miR-103a-3p	
hsa-miR-210-3p		hsa-miR-142-3p	
hsa-miR-3065-3p		hsa-miR-148a-3p	
hsa-miR-3617-5p		hsa-miR-193a-5p	
hsa-miR-4772-3p		hsa-miR-195-5p	
hsa-miR-491-5p		hsa-miR-6721-5p	
hsa-miR-627-5p		hsa-miR-92	
hsa-miR-651-5p		hsa-miR-206	
hsa-miR-664b-3p		hsa-miR-185-5p	Grehl et al., 2021 [50]
hsa-miR-766-3p		hsa-miR-320a-3p	
hsa-miR-122b-5p	Grehl et al., 2021 [50]	hsa-miR-320b	
hsa-miR-144-5p		hsa-miR-320c	
hsa-miR-193b-3p		hsa-miR-320d	
hsa-miR-29b-3p		hsa-miR-4742-3p	
hsa-miR-454-3p		hsa-miR-125b-5p	Li et al., 2021 [57]
hsa-miR-144-3p	Li et al., 2021 [57]	hsa-miR-142-5p	
hsa-miR-183-5p		hsa-miR-16-2-3p	
hsa-miR-18a-5p		hsa-miR-15b-5p	Tang et al., 2020 [46]
hsa-miR-18b-5p		hsa-miR-486-5p	
hsa-miR-20a-5p		hsa-miR-15a-5p	Fayyad-Kazan et al., 2021 [55]
hsa-miR-3613-5p		hsa-miR-19a-3p	
hsa-miR-146a-5p	Tang et al., 2020 [46]	hsa-miR-19b-3p	
hsa-miR-181a-2-3p		hsa-miR-23a-3p	
hsa-miR-17-5p	Fayyad-Kazan et al., 2021 [55]	hsa-miR-194-5p	
		hsa-miR-146a-3p	Donyavi et al., 2021 [60]

## Data Availability

Publicly available datasets were analyzed in this study. These data were deposited in the GEO database under accession number GSE169504.

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
