# Peer review of "Gene Network Analysis of the Transcriptome Impact of SARS-CoV-2 Interacting MicroRNAs in COVID-19 Disease"

_ijms, 2022, doi:10.3390/ijms23169239_

Round 1

Reviewer 1 Report

Gene Network Analysis of the Transcriptome Impact of MicroRNAs in COVID-19 disease

In this review authors tried to summarise the findings of the studies that looked at microRNAs associated with COVID-19. They took the differentially expressed microRNA target genes, identified the gene clusters with STRING and looked at the functional enrichment of the genes in the prominent clusters. This is an interesting exercise. Unfortunately, this does not qualify as a review. The manuscript is extremely poorly written and needs to be rewritten to be accepted even as a regular manuscript. To this manuscript to be accepted as a review it needs to be restructured and rewritten completely. See the comments below for pointers.

1.     Although the journal editor sent this to me as a “review” this manuscript is not written as a review.

(a)   Title has no synthesis: Neither a conclusion nor a statement but the description of what the authors tried to do.

(b)  Introduction lacks general introduction to microRNAs before started talking about COVID-19 associated ones

(c)   There is no structure overall for a review: The manuscript is written as introduction, results, methods, and discussion. 

2.     Most of the claims in the Micro RNA section in the introduction (lines 64-82) are unsupported by published literature. Authors went on to say "Many bioinformatic studies" in line 72 and move all citations to a Supplementary file. This is not acceptable. The studies that you make use of deserved to get a proper citing.

3.     All figures are extremely poor quality and not legible assess them. Also see comment for Figure 1 below.

4.     Figure 1 has multiple issues: 

(a)   Poor quality with blurry text 

(b)  No description in the figure caption. For example, N - number of publications? is up arrow means upregulated? 

(c)   28+26=54 not 44? Which numbers are correct? 

(d)  Cite the references for the tools used? e.g. miRWalk 3.0 not cited the original work anywhere.

5.     The manuscript is poorly written as I mentioned above. In many places it is not clear what the authors intended. I am  providing two examples to illustrate what I mean

e.g.1: From lines   95-98: It is not clear what the authors tried to say. Please break this long sentence in to multiple sentences. Also provide enough details for the reader to understand what you intended. Also 28+26=54 not 58

e.g.2: What did the authors mean by experimental validation? Confirming whether a in silico predicted miRNA is expressed as an RNA? Or processed into a mature miRNA? or whether it is DE upon SARS CO-V2 infection? Please provide enough and clear information.

6.     Supplementary File 3 and 4: How the targets and the microRNAs pair? Every miRNA has a single target except hsa-miR-766-3p which has all the rest listed in the opposite column? Is this correct.

What is the purpose of Supplementary File 5? Please provide a high resolution .pdf or .eps file as a supplementary figure. 

Author Response

Please see the attachment (for Reviewer 1).

Reviewer 2 Report

This is a clear and concise analysis of miRNA associated with Covid-19 infection, specifically at the tissue/organ levels. The authors have assessed and bioinformatically interrogated and interpreted public data from a number of studies (44- both in silicon and bench experiments) and cross referenced differentially expressed gene sets. 

I believe this study to be valuable to the scientific community in that it nicely packages data sets to date in a informative manner. The manuscript is well written and presented. The study is comprehensively supported by well referenced work.

Only point to make, is that I would like to see a graphical abstract highlighting the primary pathways involved which the authors have identified in their study. This would serve as an easy to follow reference and bench mark figure for the reader.

Author Response

We thank the reviewer for his thoughtful comments and efforts toward improving our manuscript.

As suggested, we have updated figure 1 and highlighted the pathways targeted by the deregulated microRNAs.

Reviewer 3 Report

The present manuscript is aimed at characterizing the biological impact of the microRNAs associated with COVID-19 disease by analyzing their potential impact on five distinct tissue transcriptomes.

To this end, the Authors identify a set of 58 (26 downregulated and 28 upregulated) microRNAs potentially targeting both SARS-CoV-2 (derived from in silico studies) and human (derived from functional studies) genes and try to provide a global systematic look into the biological impact of the microRNAs associated with COVID-19 to improve the understanding of COVID-19 multi-organ physiopathology.

1) Apparently, the "in silico" studies included in Supplementary File 1 are related only to miRNAs with "Predicted Targets on SARS-CoV-2 genome".

Therefore, the comparison of the "in silico" with the "functional" studies found in Supplementary file 3 should highlight the human miRNAs potentially targeting SARS-CoV-2 genes.

The Authors should explain better this aspect as well as the rationale behind the link between these miRNAs and the potential target human genes (Supplementary Files 3 and 4).

2) The size and resolution of the Figures 2, 3, 4, 5 and 6 included in the main text as well as of the Figure shown in Supp File 5 (Down-regulated genes targeted by the up-regulated microRNAs) should be increased because it is not possible to read the names of the genes.

Author Response

Reviewer 3.

The present manuscript is aimed at characterizing the biological impact of the microRNAs associated with COVID-19 disease by analyzing their potential impact on five distinct tissue transcriptomes.

To this end, the Authors identify a set of 58 (26 downregulated and 28 upregulated) microRNAs potentially targeting both SARS-CoV-2 (derived from in silico studies) and human (derived from functional studies) genes and try to provide a global systematic look into the biological impact of the microRNAs associated with COVID-19 to improve the understanding of COVID-19 multi-organ physiopathology.

We thank the reviewer for his thoughtful comments and efforts toward improving our manuscript.

1) Apparently, the "in silico" studies included in Supplementary File 1 are related only to miRNAs with "Predicted Targets on SARS-CoV-2 genome". 

Therefore, the comparison of the "in silico" with the "functional" studies found in Supplementary file 3 should highlight the human miRNAs potentially targeting SARS-CoV-2 genes.

The Authors should explain better this aspect as well as the rationale behind the link between these miRNAs and the potential target human genes (Supplementary Files 3 and 4).

We apologize for the confusion we have created; indeed, we intended to focus on SARS-COV-2-interacting microRNAs; since some of these microRNAs also (putatively) target human genes involved in Covid-19 pathophysiology (ACE2, TMPRS2, etc.), we felt the need to emphasize this. We have rewritten these paragraphs, and, hopefully, now the focus of our work is much clearer.

The list of target genes provided in supplementary file 3 represents predicted (miRWalk3.0) target genes

The list of target genes provided in supplementary file 4 represents the differentially expressed genes, putative targets of microRNAs.

2) The size and resolution of the Figures 2, 3, 4, 5 and 6 included in the main text as well as of the Figure shown in Supp File 5 (Down-regulated genes targeted by the up-regulated microRNAs) should be increased because it is not possible to read the names of the genes.

We are grateful for having the opportunity to provide new, more legible versions of figures 2-6.

Reviewer 4 Report

The findings appear to be interesting. Specific points that the authors need to address are as follows:

1. The roles of various identified DEGs should be validated by overexpression/deletion studies.

2. A limited in vivo study will greatly increase the impact of the findings.

3.  The role of PI3K/Akt inhibitor in available models of Covid-19 can be evaluated.

4. The limitations associated with the study should be discussed.

5. Proper statistical analysis should be conducted for all the figures.

6. Typographical errors were found throughout the manuscript and should be corrected.

Author Response

Please see the attachment (for reviewer 4).

Round 2

Reviewer 1 Report

none

Author Response

We thank the reviewer for his evaluation.